# Model of the Origin of a Ciguatoxic Grouper (*Plectropomus leopardus*)

**DOI:** 10.3390/toxins15030230

**Published:** 2023-03-21

**Authors:** Michael J. Holmes, Richard J. Lewis

**Affiliations:** Institute for Molecular Bioscience, The University of Queensland, Brisbane 4072, Australia; m.holmes@imb.uq.edu.au

**Keywords:** ciguatera, ciguatoxin, *Gambierdiscus*, *Plectropomus*, *Ctenochaetus*, *Acanthurus*, surgeonfish, turf algae, grouper, coral trout, Great Barrier Reef

## Abstract

Published data were used to model the transfer of ciguatoxins (CTX) across three trophic levels of a marine food chain on the Great Barrier Reef (GBR), Australia, to produce a mildly toxic common coral trout (*Plectropomus leopardus*), one of the most targeted food fishes on the GBR. Our model generated a 1.6 kg grouper with a flesh concentration of 0.1 µg/kg of Pacific-ciguatoxin-1 (P-CTX-1 = CTX1B) from 1.1 to 4.3 µg of P-CTX-1 equivalents (eq.) entering the food chain from 0.7 to 2.7 million benthic dinoflagellates (*Gambierdiscus* sp.) producing 1.6 pg/cell of the P-CTX-1 precursor, P-CTX-4B (CTX4B). We simulated the food chain transfer of ciguatoxins via surgeonfishes by modelling *Ctenochaetus striatus* feeding on turf algae. A *C. striatus* feeding on ≥1000 *Gambierdiscus*/cm^2^ of turf algae accumulates sufficient toxin in <2 days that when preyed on, produces a 1.6 kg common coral trout with a flesh concentration of 0.1 µg/kg P-CTX-1. Our model shows that even transient blooms of highly ciguatoxic *Gambierdiscus* can generate ciguateric fishes. In contrast, sparse cell densities of ≤10 *Gambierdiscus*/cm^2^ are unlikely to pose a significant risk, at least in areas where the P-CTX-1 family of ciguatoxins predominate. The ciguatera risk from intermediate *Gambierdiscus* densities (~100 cells/cm^2^) is more difficult to assess, as it requires feeding times for surgeonfish (~4–14 days) that overlap with turnover rates of turf algae that are grazed by herbivorous fishes, at least in regions such as the GBR, where stocks of herbivorous fishes are not impacted by fishing. We use our model to explore how the duration of ciguatoxic *Gambierdiscus* blooms, the type of ciguatoxins they produce, and fish feeding behaviours can produce differences in relative toxicities between trophic levels. Our simple model indicates thresholds for the design of risk and mitigation strategies for ciguatera and the variables that can be manipulated to explore alternate scenarios for the accumulation and transfer of P-CTX-1 analogues through marine food chains and, potentially, for other ciguatoxins in other regions, as more data become available.

## 1. Introduction

Ciguatera is a form of food poisoning caused by eating normally edible species of warm water fish that have accumulated ciguatoxins (CTX) through their diet [1]. It is an uncommon but underreported disease from the Great Barrier Reef (GBR) in Queensland, Australia. The GBR is ~2300 km long and consists of ~3000 separate reefs, ~600 mainland islands, and ~300 coral cays. Many demersal and some pelagic fish species from the GBR have caused ciguatera [1,2,3,4], with written records of poisonings from at least the early 20th-century [5,6,7]. Coral trout is the local name for groupers belonging to species in the genera *Plectropomus* (Bloch 1790) and *Variola* Swainson 1839 (Appendix A), which are often implicated in causing ciguatera outbreaks from the GBR [1,8]. However, this association probably reflects the high tonnages caught from the GBR rather than any species-specific increased risk, as coral trout are highly desirable food fishes targeted by both commercial and recreational fishers. The common coral trout (*P. leopardus* (Lacépède 1802)) is the main target species of the commercial, coral reef, fin-fish fishery on the GBR [9] and consists of a single stock along the east coast of Queensland [10]. This stock is currently considered sustainably fished on the GBR [9], with the population level in 2020 estimated to be ~59% of the unfished spawning biomass [11]. As part of measures to protect the stock, State fisheries legislation prohibits commercial and recreational fishers from taking common coral trout <38 cm total length (~0.7–0.8 kg wet weight, [12]), with the average size harvested by commercial fishers being ~1.6 kg [9]. Campbell et al. [10] assessed the annual Queensland harvest at ~829 tonnes by commercial fishers and ~171 tonnes by recreational fishers, and these figures combined suggest that >600,000 fish are harvested annually from the GBR [13].

Pacific-ciguatoxin-1 (P-CTX-1), which is also known as CTX1B [14], and its less-toxic 54-deoxy analogues (P-CTX-2, and -3, [15]) are the major ciguatoxins so far found from ciguateric coral trout from the GBR [16]. Ciguatoxins are produced by benthic dinoflagellates that belong to the genera *Gambierdiscus* Adachi and Fukuyo 1979, and *Fukuyoa* Gómez, Qiu, Lopes, and Lin 2015 [14]. Eighteen species of *Gambierdiscus* and four species of *Fukuyoa* have been described to date [14], with six *Gambierdiscus* and one *Fukuyoa* species so far found from the GBR (reviewed by Holmes et al. [13]). Globally, the highest cellular production of ciguatoxins was reported from *G. polynesiensis* Chinain and Faust 1999 by HPLC-MS/MS and cell-based and immuno-assays, and from *G. excentricus* Fraga 2011 by cell-based and immuno-assays [17,18,19,20,21], but these species have not yet been identified from the GBR. Ciguatoxins have been extracted from a cultured *Gambierdiscus* clone from the GBR [22] and ciguatoxin-like activity has been reported from cultures of *G. lapillus* Kretzschmar, Hoppenrath, and Murray 2017, *G. lewisii* Kretzschmar, Larsson, Hoppenrath, Doblin, and Murray 2019, and *G. holmesii* Kretzschmar, Larsson, Hoppenrath, Doblin, and Murray 2019, isolated from the GBR [23].

The precursors of P-CTX-1, -2, and -3, are P-CTX-4A and its chemically more stable stereoisomer P-CTX-4B (also known as CTX4A and CTX4B, respectively) [14,24,25,26]. As P-CTX-1, -2, and -3 are the major ciguatoxins found in demersal and pelagic ciguateric fishes from the east coast of Australia [16,27,28], some species/strains of east coast Australian *Gambierdiscus* (and possibly *Fukuyoa*) must produce significant concentrations of P-CTX-4A/B, which are then bio-transformed in food chains to P-CTX-1 [29]. *Gambierdiscus polynesiensis* is the only species so far confirmed to produce these ciguatoxin precursors [14,17,30,31,32,33], although there are indications that other *Gambierdiscus* and *Fukuyoa* species may be able to produce P-CTX-1-related analogues [21,34]. The major CTX analogues so far found from French Polynesian *G. polynesiensis* belong to the P-CTX3C (CTX3C) family of toxins [14,32,33], which have a different structural backbone (“E”-ring) to the P-CTX-1 congeners [26] and cannot be converted to P-CTX-1 in marine food chains. P-CTX3C analogues have not yet been found from ciguateric fish or *Gambierdiscus* from Australia; however, this may be due to lack of studies. In Japan, ciguateric reef fish are also predominantly contaminated with the P-CTX-1 family of toxins [35,36,37,38,39,40], but P-CTX3C analogues have been detected from spotted knifejaw (*Oplegnathus punctatus* (Temminck and Schlegel 1844)) from the Miyazaki coast and from red bass *(Lutjanus bohar* (Forsskål 1775)) from Minamitorishima [35].

*Gambierdiscus* and *Fukuyoa* species grow as epiphytes on a range of substrates, with most populations quantified based upon the wet weight of macroalgae. However, macroalgae typically have low standing biomass on mid- and offshore reefs of the GBR [41,42], with no evidence of any trend for macroalgae increasing across Indo-West Pacific coral reefs [43]. Turf algae are the more likely substrate supporting benthic dinoflagellate populations in the ciguateric food chain of much of the GBR (reviewed by Holmes et al. [13]). Until recently, turf algae have been an overlooked source for the origin of ciguateric benthic dinoflagellates [13,29,44,45,46,47,48]. On the GBR, ciguatoxins likely enter the human food chain from herbivorous fishes grazing ciguatoxin-producing *Gambierdiscus* on turf algae, which are, in turn, preyed upon by meso-predatory fishes such as coral trout [13,29,44]. The surgeonfishes (Acanthuridae) comprise a major group of the herbivorous/detrititivorous fishes found on the GBR (>25 species), with species of *Ctenochaetus* Gill 1884 and the dusky/brown surgeonfish (*Acanthurus nigrofuscus* (Forsskål 1775)) abundant and often dominant on mid- and outer-shelf reefs [49]. The lined surgeonfish (*C. striatus* (Quoy and Gaimard 1825)) is often one of the most dominant of these species [50,51] and is a major vector of ciguatoxins in the Pacific Ocean [52] in addition to being the cause of ciguatera where it is eaten [53,54,55,56]. It feeds by using its bristle-like teeth to comb turf algae for detritus and then brushes its teeth against a lightly keratinized structure in the upper jaw called a retention plate, described as a “dustpan-and-brush” feeding mechanism, where the retention plate acts as the “dustpan” [57,58]. The focus of feeding by *C. striatus* on turf algae has been disputed (detritovore/herbivore), but the current consensus appears to consider it a detritivore that removes mostly only small amounts of turf algae [57,58]. In contrast, the similar sized and often visually similar *A. nigrofuscus* feeds by cropping or ripping turf algae from its substrate [57,59,60]. Both methods of feeding result in the ingestion of epiphytic benthic dinoflagellates from turf algae, with microscopic observations and recent DNA sequence analysis of stomach contents from *C. striatus* confirming the presence of dinoflagellates [61,62]. Grazing by *C. striatus* is temperature- (seasonal) and sediment-dependant, with *C. striatus* sometimes rejecting (spitting out) bites with high loads of fine sediments, whereas feeding by *A. nigrofuscus* appears to be independent of sediment load [57,58,63].

In this paper, we model the hypothetical food chain transfer of CTX from *Gambierdiscus* through surgeonfishes to the common coral trout (*P. leopardus*) (Figure 1) to obtain a quantitative estimate of the density of a *Gambierdiscus* bloom to produce a ciguateric fish on the GBR. This model allowed us to assess the influence of ciguatoxic *Gambierdiscus* bloom duration, the type of ciguatoxins produced, and fish feeding behaviours on the relative toxicities between trophic levels. We use *C. striatus* as our model for the transfer of ciguatoxins by surgeonfishes to common coral trout but, where possible, include comparisons with *A. nigrofuscus* as an alternate vector. This contribution builds upon previous modelling for the production of ciguateric pelagic fish (Spanish mackerel) from the east coast of Australia [64] and conceptual models for ciguateric food chains on the GBR, as well as models for the reduction in CTX concentrations in fishes through growth and depuration [13]. Our model synthesizes research on ciguatera, fisheries, and coral reef ecology to produce quantitative estimates for the flow of CTX through food chains. This model can be further refined as new data are acquired and allows testing of interactions between variables to be explored to match specific scenarios of interest.

Our modelling assumes the transfer of ciguatoxins across a three-trophic-level food chain (Figure 1); however, alternate food chains such as through cryptobenthic fishes or invertebrates likely also contribute to ciguatera risk [13].

## 2. Results and Discussion

### 2.1. Framework for Model Development

Bomber [65], Caire et al. [66], and Lobel et al. [67] were probably the first to quantify *Gambierdiscus* populations based on surface area rather than wet weight of substrates. Subsequently, underwater vacuum devices were used to explore *Gambierdiscus* populations based upon the surface area of turf algae grazed by surgeonfish [44,68]. An interesting alternative was developed by Loeffler et al. [69] for monitoring benthic dinoflagellate populations on turf algae growing on tiles. However, recent studies quantifying benthic dinoflagellate populations by surface area have used mostly mesh screens deployed for 24 h near the benthos [45,47,48,70,71,72,73,74]. Parsons et al. [48] suggested that populations of *Gambierdiscus* on mesh screens correlated poorly with populations on macroalgae, although Tester et al. [71] have shown otherwise. However, for the GBR, it is more important to determine how such screen counts correlate with populations on turf algae, as this methodology could greatly simplify the assessment of *Gambierdiscus* populations grazed by surgeonfishes. In the future, a method that quantifies ciguatoxins and *Gambierdiscus* per unit area would be the most useful for assessing the risk for the production of toxic fish [48].

The maximum density of *Gambierdiscus* so far reported from artificial substrates is 350 cells/cm^2^ [70], although regional maximum densities are more typically ~100 cells/cm^2^ [45,70,71,72]. However, most *Gambierdiscus* densities reported from screens are in the order of ~0.1—10 cells/cm^2^, which may be analogous to the widespread but low concentrations of *Gambierdiscus* that are found on many macroalgal substrates from reefs around the world [14]. The detection of *Gambierdiscus* from many sites using the mesh screen assay of Tester et al. [70] has demonstrated that considerable numbers of some *Gambierdiscus* species must often be in the water column (tychoplanktonic), providing a source of cells to seed populations on turf algae and other substrates. Given the generally slow growth rates of *Gambierdiscus* species [17,19,75,76,77,78], including isolates from the GBR [79,80], the contribution from cell division to screen populations over the 24 h deployment time of the assay would likely be minimal and at most <2-fold. The maximum density of *Gambierdiscus* that we have observed in cultures grown on the flat surface of 50 mL tissue culture flasks [81,82] is ~240 cells/cm^2^, with cells often initially growing in clumps. Only later as the population density increases do cells become more uniformly distributed over the bottom of the flask (unpublished observations). In the quiescent environment of a culture flask, this likely occurs by dividing cells mostly not moving from their initial attachment point, with occasional free-swimming cells forming new colonies. The screen assay incorporates the surface area from the three-dimensional mesh surface [70,71], which would be an underestimate of densities based only on an equivalent two-dimensional area (plan view). The three-dimensional structure of turf algae provides considerably more surface area for *Gambierdiscus* to attach and grow than screens and increases the complexity of trying to model the transfer of CTX to surgeonfish grazing them. Our model of a GBR ciguatera food chain is based upon surgeonfish grazing an essentially two-dimensional surface layer, although this is a simplification of the diversity and structure of turf algae [83].

We modelled the hypothetical transfer of ciguatoxins from *Gambierdiscus* epiphytic on a turf algae layer at surface (plan view) densities of ≤10, 100, and 1000 cells/cm^2^ to span the globally reported cell densities from surfaces (mostly screens). We based our model on these cells producing 1.6 pg of P-CTX-4B/cell, which was the same theoretical toxin concentration used for our recent modelling of the contamination of pelagic Spanish mackerel (*Scomberomorus commerson* (Lacépède 1800)) by *Gambierdiscus* from Platypus Bay on the east coast of Queensland (a non-coral reef habitat south of the GBR) [64]. This CTX concentration is 2.7-fold greater than the maximum known combined cellular concentrations of P-CTX-4A and -4B of 0.6 pg/cell produced by cultured *G. polynesiensis* [33]. The only known CTX concentration from *Gambierdiscus* isolated from the GBR was extracted from a cultured clone (NQ2/7) isolated from Arlington Reef, ~40 km northeast of the city of Cairns, with ciguatoxin quantified using mouse bioassay [22]. The ciguatoxin concentration (mouse units/cell) produced by this clone was a similar order-of-magnitude to that extracted from the cultured WC1/1 clone isolated from Platypus Bay [22]. However, it is likely that turf algae often host a mix of *Gambierdiscus* species and/or strains containing a range of ciguatoxin concentrations.

We modelled the contamination of a 1.6 kg common coral trout (the average size of the commercially harvested fish of this species from the GBR [9]), accumulating a target flesh CTX contamination of 0.1 µg/kg P-CTX-1. This concentration is 10-fold higher than what the USFDA recommends as the safe limit for the consumption of seafood [84] and is the lowest effect level determined by Lehane and Lewis [85], and formed the basis of the USFDA compliance level for P-CTX. This flesh concentration of CTX would likely produce mild symptoms in 2 out of 10 people [85]. We calculated a total CTX burden for a 1.6 kg common coral trout by using the method of Holmes and Lewis [64], based upon a flesh (fillet) recovery weight of 50% total fish weight [86,87] and a toxin assimilation efficiency of 43% across each trophic level, with flesh contributing 10–40% of the total CTX burden [64,88]. On this basis, a 1.6 kg common coral trout would accumulate between 0.2 and 0.8 µg P-CTX-1 equivalents (eq.). from its diet to produce the modelled level of CTX contamination in the flesh of the fish (Appendix A), and this level of CTX would have originated from 1.1 to 4.3 µg P-CTX-1 eq. produced by *Gambierdiscus* at the base of a three-trophic level food chain (Appendix A). Assuming that each cell produced 1.6 pg of P-CTX-4B, this would equate to a highly ciguatoxic *Gambierdiscus* population of between 0.7 and 2.7 million cells to produce a 1.6 kg common coral trout with a flesh contamination of 0.1 µg/kg P-CTX-1 (Appendix A). Based upon our model, the 1.6 kg common coral trout would have acquired this CTX burden by preying upon one or more surgeonfish that had consumed this population of ciguatoxic *Gambierdiscus* from turf algae.

For a single surgeonfish to consume 0.7–2.7 million *Gambierdiscus* from turf algae supporting cell densities between 10 and 1000 cells/cm^2^, would require the fish to graze between 0.07 and 27 m^2^ of turf algae, assuming grazing was 100% efficient at removing these epiphytes. However, grazing efficiency is likely to be less than this, as grazing scars for *C. striatus* feeding on laboratory-prepared surfaces show apparently intact areas between the teeth lines on scraped substrates [58,59], allowing space for cells to possibly avoid being ingested. Similarly, any *Gambierdiscus* attached to turf algae below the bite of *A. nigrofuscus* could avoid being consumed, as *A. nigrofuscus* tend to nip the top of turf algae, avoiding any underlying layer of sediments [59,60,63]. The feeding mechanisms of both *C. striatus* and *A. nigrofuscus* tend not to leave size-specific grazing scars on the turf algal substrate from their bites [51], so we can only base our estimate for the area grazed by *C. striatus* on grazing scars produced in laboratory experiments [58]. We have assumed 50% efficacy for ingesting *Gambierdiscus* from turf algae based on the spacing of the teeth marks from the grazing scars left on aquaria surfaces by ~18 cm (total length) *C. striatus* [58]; however, the actual efficacy is unknown and probably varies with the composition of algal species in the turf and their aspect on the reef. We estimate the area of the grazing scar from the lower jaw of an ~18 cm fish in Figure 4c of [58] to be ~0.7 cm^2^ and have ignored the contribution from the upper jaw, as Tebbett et al. [58] suggest it contributes minimally to biting (~18% of the total bite area).

We have based our model on the bite area of an 18 cm *C. striatus* [58] because there are no other comparable data available. This is a large size fish for this species that would weigh ~102 g [12]. Under optimal feeding conditions, coral trout <2 kg consume an average of 4.6% of their body weight (corresponding to ~74 g for a 1.6 kg coral trout) [89], although feeding rates and meal size are dependent upon many factors including water temperature and predator and prey body size [89,90]. A 74 g *C. striatus* would have a total length between 16 and 17 cm [12], so it would presumably have a similar bite area to that of the 18 cm fish upon which we have based our model.

Surgeonfishes often feed over reefs in large schools, with *C. striatus* and *A. nigrofuscus* being diurnal feeders that have most feeding activity towards the middle of the day [91,92,93]. The maximum feeding rate we are aware of for *C. striatus* was recorded from Moorea (French Polynesia), with an average of 17.5 bites/min over a 9 h feeding period and >12,600 bites/day [93]. These feeding rates are much higher than those recorded from the GBR [60,63], with an average of ~10 bites/min on GBR turf algae that contain low concentrations of sediments [63]. *Acanthurus nigrofuscus* bites more often than *C. striatus*, with an average of ~20 bites/min independent of sediment concentration on the GBR [57,63] and >30 bites/min on Hawaiian reefs [94]. Our model is consistent with these data and assumes *C. striatus* grazes on GBR turf algae at an average of 10 bites/min over 9 h/day (5400 bites/day).

### 2.2. Scenario 1: 1000 Gambierdiscus/cm^2^ Epiphytic upon Turf Algae

Our modelling suggests that a 74 g *C. striatus* feeding at 5400 bites/day on turf algae hosting a bloom of 1000 *Gambierdiscus*/cm^2^ producing an average of 1.6 pg of P-CTX-4B/cell would accumulate sufficient CTX in <2 days (Table 1) to contaminate the flesh of a 1.6 kg common coral trout with 0.1 µg/kg P-CTX-1 (i.e., a mildly toxic fish). If we based our model on the maximum known combined concentrations of P-CTX-4A and -4B/cell of 0.6 pg/cell from *G. polynesiensis* [33], then it would take between 1 and 4 days for *C. striatus* to accumulate a similar level of P-CTX (Table 1). The relatively short time frame for the accumulation of CTX into surgeonfish under either scenario indicates that population levels of ~1000 cells/cm^2^ of highly ciguatoxic *Gambierdiscus* present a high risk for the development of ciguatoxic meso-predatory fishes that are the focus for many reef fisheries around the world, including the GBR. At present, our modelled cell density of 1000 cells/cm^2^ is a hypothetical scenario, as we are not aware of any reports of such high cell surface densities from the wild.

### 2.3. Scenario 2: ≤10 Gambierdiscus/cm^2^ Epiphytic upon Turf Algae

Population densities of between 0.1 and 10 *Gambierdiscus*/cm^2^ are commonly found on the mesh screen assay at many sites around the world [45,71,72]. We consider these to be sparse cell densities and explore the impact from ≤10 *Gambierdiscus*/cm^2^ on turf algae, but we do not know how comparable these densities are to those on artificial substrates. Our modelling suggests that a 74 g *C. striatus* feeding at 5400 bites/day on turf algae that hosts 10 *Gambierdiscus*/cm^2^ and produces an average of 1.6 pg of P-CTX-4B/cell would have to feed between ~1 and 5 months (Table 1) to accumulate sufficient CTX that, if preyed upon by a 1.6 kg common coral trout, would result in the grouper developing a flesh contamination of 0.1 µg/kg P-CTX-1. We do not know if highly ciguatoxic *Gambierdiscus* can be maintained on reefs for this length of time, as Liefer et al. [95] have shown that the ciguatoxicity of *Gambierdiscus* populations can vary seasonally in the Caribbean.

The median value for global densities of *Gambierdiscus* on screens appears to be ~1 *Gambierdiscus*/cm^2^ [71], although this is from a relatively small data set. Our modelling suggests that it would require ~1–4 years of grazing on turf algae with 1 *Gambierdiscus*/cm^2^ by a 74 g *C. striatus* to accumulate sufficient ciguatoxin burden to pose a risk of ciguatera if the *C. striatus* were preyed upon by higher trophic-level fishes. Ciguatoxin accumulation from these long grazing times could be wholly or partly offset by decreases through depuration processes (reviewed by Holmes et al. [13]). Therefore, sparse populations of highly ciguatoxic *Gambierdiscus* (≤~10 cells/cm^2^), are unlikely to generate ciguatoxic higher trophic-level fishes, at least in areas where the P-CTX-1 family of ciguatoxins predominate.

An additional factor that could limit the flow of CTX through the GBR food chain (Figure 1) is the feeding behaviour of our modelled predator. Coral trout are opportunistic, apex predators that prey principally on fishes [96,97,98,99], but the evidence suggests that *P. leopardus* rarely preys upon surgeonfish on the GBR, including *C. striatus* and *A. nigrofuscus* (reviewed by Holmes et al. [13]). If surgeonfish are not preyed upon by higher trophic-level fishes, any CTX burden accumulated by them is unlikely to enter the human food chain in Australia, as herbivorous reef fishes are generally not harvested for food from the GBR [100,101]. A mechanism that may facilitate predation of surgeonfish by opportunistic predators is behavioural changes (such as inactivity or swimming irregularities) induced in prey fishes that ingest toxic secondary metabolites such as those produced by benthic dinoflagellates (including ciguatoxins and maitotoxins) [13,29,102,103,104,105]. Laboratory experiments feeding *Gambierdiscus* to fishes have sometimes induced observable behavioural changes [102,103,104,105], but this was not the case in other studies [106]. However, these differences could result from the use of different *Gambierdiscus* species containing different suites of bioactive compounds and/or the use of different experimental feeding rates. Surgeonfishes may also be able to acclimate to dinoflagellate toxins over time [104], which might reduce their risk of predation and limit the flow of ciguatoxins to higher trophic-level fishes.

*Gambierdiscus* are common, low-density epiphytes on a diverse range of substrates on the GBR [107,108]. Given the ubiquitous nature of surgeonfish grazing on turf algae on the GBR [49,50,51,60], it is likely that most surgeonfishes that graze these turf algae continuously ingest sparse populations of *Gambierdiscus* (and other benthic dinoflagellates) and their associated secondary metabolites. As the probability of surgeonfishes from the GBR being preyed upon by coral trout is low (reviewed by Holmes et al. [13]), it suggests that the continuous, long-term ingestion of benthic dinoflagellate secondary metabolites from low-density populations of *Gambierdiscus* on the GBR have no major detrimental impact on surgeonfish behaviour. This is further support for our hypothesis that surgeonfish feeding on sparse *Gambierdiscus* populations present a low risk for the development of ciguateric meso-predatory fishes such as coral trout on the GBR. Given that sparse *Gambierdiscus* populations are the most common scenario on reefs globally [14,71], this is likely the default situation for most marine food chains.

We are aware of only five studies in which *Gambierdiscus* cells were directly fed to fish [102,103,104,105,106], with only the study of Clausing et al. [106] and possibly those of Davin et al. [102,103] using experimental feeding rates based upon ecologically relevant populations of *Gambierdiscus*. However, even the feeding rates used by Clausing et al. [106] may not be ecologically relevant for surgeonfish that graze on turf algae. An adult *C. striatus* feeding on low-density populations of *Gambierdiscus* (10 cells/cm^2^) at 5400 bites/day with each bite scraping 0.7 cm^2^ of turf algae could ingest ~37,800 *Gambierdiscus*/day. For our model, we have assumed a more conservative cell population ingested per day (~18,900 cells) by incorporating an efficiency of only 50% for consuming *Gambierdiscus* from a bite of turf algae and ignored any contribution of the upper jaw to the bite. Additionally, our modelled daily feeding rate is less than half that recorded for *C. striatus* in French Polynesia [93]. If we modelled feeding by our alternate vector (*A. nigrofuscus*), we would have to use a bite rate of at least twice that we have used for *C. striatus* [63,94].

Clausing et al. [106] fed 89 cells/g fish body weight/day to juvenile 6–30 g surgeonfish (*Naso brevirostris* (Cuvier 1829)) that typically feed on macroalgae. This corresponds to an initial feeding rate of between ~272 and 1033 *Gambierdiscus*/day, which increased to >2000 cells/day by the end of the 16-week experiment, with some juvenile surgeonfish therefore ingesting >2.8 million *Gambierdiscus* over the duration of the experiment (based on Table 2 in Clausing et al. [106]). If we applied the feeding rate of 89 *Gambierdiscus*/g body weight of fish used by Clausing et al. [106] to our hypothetical 74 g *C. striatus*, it would correspond to an ingestion rate of only 6586 cells/day. However, our modelling suggests that feeding rates of approximately three times this would correspond to surgeonfish feeding only on sparse populations of *Gambierdiscus*. Much greater feeding rates may be more relevant for the trophic transfer of CTX, especially where the P-CTX-1 family of toxins are the principal cause of ciguatera. We suggest that likely ingestion rates of *Gambierdiscus* in the wild should be incorporated as a variable in future feeding experiments, which may also resolve any contribution of cell density and the production of secondary metabolites to inducing behavioural changes in herbivorous fishes. Unfortunately, it is not possible to estimate the number of *Gambierdiscus* fed to surgeonfish (*Acanthurus bahianus* Castelnau 1855, and *A. chirurgus* (Bloch 1787)), by Magnelia et al. [104], or to juvenile (~11 g) mullet (*Mugil cephalus* Linnaeus 1758), by Ledreux et al. [105]. In comparison, Davin et al. [102] fed ~10,000–30,000 *Gambierdiscus* once only to 2–4.5 g bluehead wrasse (*Thalassoma bifasciatum* (Bloch 1791)), and Davin et al. [103] fed ~20,000–200,000 *Gambierdiscus* once only to freshwater largemouth bass (*Micropterus salmoides* (Lacépède 1802)).

A 74 g *C. striatus* is likely to have a total length between 16 and 17 cm [12], which is an adult fish that is several years old [109,110,111]. However, the generally higher mortality suffered by juvenile fishes in the wild may facilitate the transfer of CTX through reef food chains by these younger fish. Although *C. striatus* are common on the GBR, mass recruitment events have not been observed and appear to be rare [111]. However, on some Pacific reefs, mass recruitment of *C. striatus* and other surgeonfishes occurs periodically [111,112,113], which could facilitate the trophic transfer of ciguatoxins through large numbers of juvenile fish to opportunistic predators such as groupers. Large schools of recruits (juvenile fish) of up to 5000 *C. striatus* have been observed in American Samoa, which graze over reefs in a short-lived pulse, with most being eaten by predators [111]. However, it is also possible that juvenile fishes rapidly depurate ciguatoxins [105,114], which could be a mechanism that limits the trophic transfer of CTX from juvenile fishes to their predators. Future research should clarify differences in toxin accumulation and depuration between juvenile and adult fishes, as this has a major impact on our understanding of the flow of toxins between trophic levels. This is especially important for the herbivorous/detritivorous species that likely ingest benthic dinoflagellates over much of their life cycle and are the presumed vectors of CTX to larger carnivorous species. Ontogenetic changes in diet are common in fishes [115,116], which could limit CTX transfer in some life stages of fishes in the wild.

### 2.4. Scenario 3: 100 Gambierdiscus/cm^2^ Epiphytic upon Turf Algae

A cell density of ~100 *Gambierdiscus*/cm^2^ is consistent with several site maxima reported from around the world using the 24 h screen assay [70,71]. Our modelling suggests that a 74 g *C. striatus* feeding at 5400 bites/day on turf algae with 100 *Gambierdiscus*/cm^2^ producing an average of 1.6 pg of P-CTX-4B/cell would have to feed between ~4 and 14 days (Table 1) to accumulate sufficient ciguatoxin that, if preyed upon by a 1.6 kg common coral trout, would result in a grouper with a flesh contamination of 0.1 µg/kg P-CTX-1. This relatively short time frame suggests that cell densities of ~100 cells/cm^2^ could be a risk factor for the development of ciguateric groupers. However, this risk likely varies with factors that affect trophic transfers along food chains, such as rates of herbivore grazing.

Coral reef fish herbivores tend to concentrate their grazing on turf algae in habitats where the turfs are most productive [51,117,118]. A recent meta-analysis of turf algae productivity on coral reefs [118] suggests that herbivore grazing in the most productive areas for turf algae growth can result in a turnover of these algae in <5 days (herbivore grazing is not limited to only surgeonfishes). Since *Gambierdiscus* are slow-growing dinoflagellates with generation times often ≥4days (reviewed by Holmes et al. [13]), these turnover rates would reduce the likelihood that high densities of *Gambierdiscus* would develop on turfs. Even if a bloom should begin to develop due to patchy grazing not cropping some turf areas [119], fast turnover rates of <5 days would cap CTX accumulation by individual herbivores. In contrast, in the less productive habitats for turfs, the standing algal biomass tends to increase, and modelling by Tebbett and Bellwood [118] suggests that herbivore turnover rates plateau at ~4%/day, indicating possible turnover times for these less-productive areas in the order of ~25 days. This slow turnover rate may only partially control *Gambierdiscus* growth and, therefore, CTX flow into surgeonfish from highly ciguatoxic *Gambierdiscus* densities of ~100 cells/cm^2^, given our modelled estimate of the feeding time required (~4–14 days) for a single *C. striatus* to accumulate a sufficient CTX burden that, if preyed upon, would produce a mildly ciguateric 1.6 kg common coral trout. However, grazing turnover rates of ~25 days likely limit the flow of CTX from sparse, highly-ciguatoxic *Gambierdiscus* densities of ~≤10 cells/cm^2^, given our modelled estimate for a feeding time of ~1–5 months for a density of 10 cells/cm^2^ (Table 1).

Our analysis focuses on the GBR, where productivity and turnover of algal turfs from coral reefs are well studied [118]. It remains to be seen how applicable these turnover rates are to coral reefs in other regions, especially where herbivorous fish stocks have been depleted by fishing. On the GBR, herbivorous fish stocks remain high, as they are generally not harvested for food [100,101]. We have suggested that significant reductions in herbivorous fish populations could be a mechanism that increases ciguatera if it reduces grazing on benthic dinoflagellate substrates and allows time for the development of ciguatoxic *Gambierdiscus* blooms [13]. Heavy fishing pressure that reduces herbivorous fish populations is also likely to have depleted predatory fish populations, which may be a concentrating mechanism that funnels ciguatoxins to higher trophic levels.

Conditions that promote the growth of *Gambierdiscus* populations are likely not the same as those that promote turf algae productivity, as turf communities show a general lack of photo-inhibition that likely underpins their high productivity at the shallowest water depths, with most production reliant on the apical portion of filaments, i.e., above the sediment layer and exposed to higher water motion and light [118]. In contrast, *Gambierdiscus* species appear to be well adapted to low light conditions [75,120,121], with the highest population densities at around the 10 m depth on screens in the Canary Islands [73], although Loeffler et al. [69] found no significant differences between populations at 10 m and 23 m depths at sites in the Caribbean. However, what is often missing from environmental studies on *Gambierdiscus* populations is the consideration that benthic dinoflagellate populations may also be controlled by herbivory and that the populations measured at any one time are likely the result of interactions between growth rates that vary with environmental gradients, offset against gradients of herbivory. Loeffler et al. [69] were the first to investigate this, showing that *Gambierdiscus* populations in the Caribbean increased when their substrates were caged to prevent access by grazers. These types of experiments may help unravel the relative importance of top-down and bottom-up ecological processes that control the flow of ciguatoxins into food chains [13].

### 2.5. Accumulation of CTX into Common Coral Trout (P. leopardus)

Ciguatera is a sporadic but generally low-incidence disease from the GBR, with an average (± 1 standard deviation) of 4.5 ± 5.4 and 4.2 ± 3.4 cases reported annually between 2017 and 2022 from the public health units of the two largest regional cities adjacent to the GBR, Cairns and Townsville, respectively [122]. Commercial fishers often catch multiple coral trout off the same localized coral reef structures (bommies) on the GBR [9]. However, multiple ciguatera outbreaks over a short period of time are rarely reported from the GBR, suggesting ciguatera on the GBR mainly occurs through localized pulsed events.

Our model for the trophic transfer of CTX to groupers on the GBR is based on a 1.6 kg coral trout preying upon a single toxic surgeonfish. This single transfer event has allowed us to estimate the minimum size of ciguatoxic *Gambierdiscus* populations required to produce a toxic grouper. However, groupers could accumulate toxin loads from repeatedly preying upon toxic surgeonfish; although, every predation event offers scope for losses that reduce the efficiency of toxin transfer. Predators such as groupers feed intermittently [90], with feeding rates for common coral trout depending upon body size and water temperature, with generally 2–4 days between meals [89]. This suggests that if groupers accumulate ciguatoxins by repeatedly feeding upon toxic prey, they likely accumulate them in a discrete manor (Figure 2). Although highly stylized, if we assume a school of surgeonfish contaminated with CTX from grazing on a short-lived bloom of highly ciguatoxic *Gambierdiscus* remained susceptible to predation over ~1 month, then a grouper repeatedly feeding every fourth day upon one of these surgeonfish carrying a similar CTX load could increase its toxin burden ~8-fold over this time (Figure 2a). In contrast, if a highly ciguatoxic bloom of *Gambierdiscus* with constant CTX cell concentration persisted over ~1 month, then daily grazing by a school of surgeonfish on this bloom might increase their CTX burden daily, resulting in a grouper feeding every fourth day on an individual from an increasingly toxic school of herbivores, possibly leading to >100-fold increase in CTX burden over this same time frame (Figure 2b). While these scenarios for developing toxin-burdens in prey and predator fish help conceptualize processes that lead to ciguateric outbreaks, variations in predator–prey responses and the homogeneity and levels of CTX being transferred produce many possible outcomes. However, the production of a grouper that would cause ciguatera would likely take longer than indicated by Figure 2 because of the time required for the metabolization and transfer of CTX from the gut into fish muscle [105,106,114,123], at least for regions where ciguatera is mostly caused by people eating fish fillets (such as the GBR).

If groupers accumulate a CTX burden by feeding on multiple contaminated prey on the GBR, it is more likely from a single pulse of toxin such as that conceptualized in Figure 2a. In contrast, in regions of the Pacific where areas remain toxic for long periods of time and fishes can accumulate very high CTX concentrations [124,125,126], the underlying processes may sometimes be more similar to Figure 2b. Interestingly, if surgeonfishes continuously grazed a persistent highly toxic *Gambierdiscus* population but acclimated to any toxic effects [104] that subsequently reduced their susceptibility to be preyed upon, then the herbivorous fishes could become more toxic than the higher trophic-level carnivores on that reef, as reported to occur in French Polynesia [55,127,128,129].

The differences in toxicity between trophic levels for herbivorous and predator fishes could also be explained by the differences in toxicity of the ciguatoxin analogues that transfer through regional marine food chains. In French Polynesia, the major ciguatoxins produced by *G. polynesiensis* are P-CTX3C analogues, with P-CTX3C as well as its suggested main food chain biotransformation product, the less toxic 2,3-dihydroxyP-CTX3C, accumulating in both herbivorous and carnivorous fishes [14,37,130,131]. That is, the same or lower relative toxicity is transferred between trophic levels. In contrast, the food chain biotransformation of P-CTX-4A/B produces a major increase in toxicity to P-CTX-2, -3, and eventually, the most toxic, P-CTX-1 [14,29,37,131,132,133], which appear to be the dominate forms of CTX toxins in Kiribati, Japan, and Australia. The impact of these differences can be conceptualized by using scenarios based upon our model for the transfer of a CTX pulse across food chains originating from a short-lived bloom of highly ciguatoxic *Gambierdiscus* (i.e., Figure 2a). Incorporating a 43% toxin assimilation efficiency [88] for the transfer of relative toxin loads from herbivorous to predator fish (Figure 3a) and then expressing these same data on an equivalent weight basis for 74 g surgeonfish and 1.6 kg coral trout show how the predator trophic level can be considerably less toxic than the herbivore level (Figure 3b). In this scenario (Figure 3b), the same relative CTX toxicity is transferred between trophic levels, as presumably occurs with P-CTX3C analogues [14]. Incorporating a five-fold increase in toxicity for the biotransformation of P-CTX-4A to P-CTX-1 [14] for the same scenario produces a considerable increase in relative toxicity at the predator trophic level (Figure 3c). We suggest that the scenario conceptualized in Figure 3b is more representative of regions where the P-CTX3C family of toxins predominate, whereas Figure 3c is likely representative of regions where the P-CTX-1 family of toxins are dominant.

The five-fold increase in toxicity for biotransformation of CTX precursors to P-CTX-1 incorporated in Figure 3c is conservative, as it corresponds to the transformation of only P-CTX-4A. In contrast, the biotransformation of P-CTX-4B to P-CTX-1 produces a 20-fold increase in toxicity [14], and P-CTX-4B is the dominant precursor so far extracted from *G. polynesiensis*, ranging between 53 and 71% of the two analogues produced by French Polynesian isolates (calculated from Table 6 in Darius et al. [33]).

The scenario conceptualized in Figure 3c relies upon the biotransformation of the less-toxic precursors to P-CTX-1 mostly occurring in the third trophic level (shown as coral trout in Figure 3c). Some support for this hypothesis is that second trophic-level parrotfish (*Chlorurus microrhinos* (Bleeker 1854)) from French Polynesia contained mostly P-CTX3C analogues and P-CTX-4A but no P-CTX-1, -2, or -3 [14]. That is, there appeared to be no conversion of P-CTX-4A to its more toxic analogues in this nominal herbivore, although the absence of P-CTX-4B is confusing. Additionally, Mak et al. [124] analysed three *Ctenochaetus striatus* from Kiribati and found that they were contaminated mostly with the less-toxic P-CTX-1 intermediates, P-CTX-2 and -3, although P-CTX-4A/B were not part of the analysis. This suggests that only partial oxidative metabolism of the precursors had occurred in these surgeonfish. Although there were species-specific differences, in general, the profiles of the P-CTX-1 family of ciguatoxins found in Kiribati surgeonfishes (*Acanthurus* and *Ctenochaetus*) tended to be dominated by the less toxic isomers P-CTX-2 and -3, relative to P-CTX-1, whereas P-CTX-1 was more common in groupers [124]. However, any accumulation of P-CTX-4A or -4B in higher trophic-level fishes consumed by people may also be metabolized to the more toxic P-CTX-1 in the human liver, as Ikehara et al. [26] have shown in vitro conversion of the precursors to P-CTX-1 by human recombinant cytochrome P450 enzyme (rhCYP3A4); however, the rate of this conversion in vivo is unknown.

The most toxic P-CTX-1 and P-CTX3C analogues are hydroxylated at the same position on the terminal “M” ring, i.e., P-CTX-1 (54-OH) and 51-hydroxyP-CTX3C (51-OH), respectively [14,130,132]. P-CTX-1 has not yet been found from *Gambierdiscus*, whereas trace amounts of P-CTX-2 and -3, as well as 51-hydroxyP-CTX3C, have been previously identified [35]. In Japan, spotted knifejaw (*Oplegnathus punctatus*) feed on invertebrates [12,37], and some of these fish have been shown to be contaminated mostly with P-CTX-4A and -4B, and only lesser amounts of P-CTX-1 [37], suggesting incomplete oxidation. Feeding on invertebrates suggests that this fish feeds at the equivalent of the third trophic level from *Gambierdiscus*. However, spotted knifejaw in Japan can also be contaminated with P-CTX3C toxins, with a report that 51-hydroxyP-CTX3C was the main analogue detected [35]. As yet, there are no studies of the production and food chain transfer of 51-hydroxyP-CTX3C, although small amounts were recently detected in moray eels [133] and red bass (*Lutjanus bohar*) [134].

### 2.6. The Minimum CTX Concentration in Gambierdiscus to Produce a Ciguateric Grouper

The variables in our simple food chain model (Appendix A) can be adjusted to explore the minimum cellular concentration of P-CTX-4A and -4B in *Gambierdiscus* that is needed to produce a mildly toxic 1.6 kg coral trout with a flesh concentration of 0.1 µg/kg P-CTX-1 (estimated minimum adverse effect level). For surgeonfish grazing for 32 days on turf algae that hosts a bloom of 100 *Gambierdiscus*/cm^2^ (consistent with some regional maximum densities detected on benthic screens, [71]), our model suggests that 0.2–0.7 pg P-CTX-4 per *Gambierdiscus* cell would be required (Appendix A). This range of concentrations appears realistic given that 21 of the 30 strains of *G. polynesiensis* extracted by Darius et al. [33] contained P-CTX-4/cell concentrations ≥0.2 pg/cell (based upon P-CTX3C eq.); however, the modelled upper limit (0.7 pg/cell) is greater than the maximum known concentration of 0.6 pg P-CTX-4/cell [33]. Given this toxin range, it is not surprising that most ciguateric groupers caught on the GBR are only mildly toxic. Our model has the potential to explore scenarios for the food chain transfer of CTX from other regions, including where CTX analogues other than the P-CTX-1 family of toxins dominate.

Our model focuses on surgeonfishes as the vector between ciguatoxic *Gambierdiscus* epiphytic on turf algae and coral trout on the GBR. However, CTX could also be transferred through invertebrates and other grazers of turf algae, such as damselfishes [13]. Damselfishes (Pomacentridae) are a diverse group of fishes that constitute a large component of the fish fauna of coral reefs, with a wide range of feeding preferences, including some that graze turf algae [135,136]. Damselfishes constitute a considerable proportion of the prey of coral trout on the GBR [99], and low concentrations of CTX have been recently detected from a single damselfish from a Caribbean reef [137], making damselfishes worthy of further study as possible alternate vectors for the trophic transfer of CTX to predatory fishes in the Caribbean Sea and, potentially, the Pacific Ocean.

Our estimate for the population of ciguatoxic *Gambierdiscus* that is required to produce a mildly toxic coral trout on the GBR is <10-times our estimate to produce a mildly toxic Spanish mackerel (*Scomberomorus commerson*) from Platypus Bay, a small bay to the south of the GBR [64]. This difference arises because groupers are typically much smaller than Spanish mackerel and probably need to ingest less toxin to produce the same level of muscle contamination, and our assumption that toxin is transferred across three trophic levels on the GBR, but across four trophic levels in Platypus Bay [64]. If *Gambierdiscus* population density were the main determining factor for the production of ciguateric higher trophic-level fish along the east coast of Australia, then it would be reasonable to assume that the chance of producing a ciguateric grouper should be greater than the production of ciguateric Spanish mackerel (all other factors being equal). However, the available data suggest that Spanish mackerel cause more ciguatera cases than coral trout in Queensland [1]; however, the epidemiological data are incomplete. This supports our suggestion that the risk for development of ciguateric fishes is more complex than merely the size of *Gambierdiscus* populations and their capacity for production of CTX, and that ciguatera risk also depends upon differences in the ecology and exploitation of marine food webs between reefs and regions.

## 3. Conclusions

Our modelling suggests that *Gambierdiscus* densities ≤10 cells/cm^2^ epiphytic on turf algae are unlikely to represent a significant risk for the development of ciguateric coral trout on the GBR and possibly other regions where the P-CTX-1 family of CTX predominate. In contrast, cell densities in the order of 1000 cells/cm^2^ of highly ciguatoxic *Gambierdiscus* likely pose a major risk for development of ciguateric groupers. Cell densities of ~100 cells/cm^2^ on turf algae are harder to assess for ciguatera risk, as this will depend upon a range of factors that impact the transfer of CTX across trophic levels. These include the intensity of grazing on turf algal substrates that support the growth of *Gambierdiscus*, and the presence and concentrations of secondary metabolites that, when consumed, may increase the probability of predation of vectors between toxin production and the fish that causes ciguatera when eaten by humans. Our modelling suggests that 0.2–0.7 pg P-CTX-4/cell is the minimum *Gambierdiscus* concentration required to allow surgeonfish grazing over 32 days on turf that supports a bloom of 100 *Gambierdiscus*/cm^2^ to contaminate the flesh of a 1.6 kg grouper with 0.1 µg/kg P-CTX-1, the estimated minimum adverse effect level. Finally, we suggest that differences in the toxicity of ciguatoxin analogues, and/or surgeonfish acclimating to the toxic effects of *Gambierdiscus* secondary metabolites, could explain regional differences in the relative toxicities of herbivorous and predatory fish trophic levels. Our modelling framework has the potential to be used to explore a range of scenarios for the flow of CTX through marine food chains, including food chains where other CTX analogues dominate.

## 4. Materials and Methods

Our modelling uses previously published data to quantify the flow of ciguatoxins (CTX) across three trophic levels of a hypothetical marine food chain on the Great Barrier Reef (GBR), Queensland, Australia, an ecosystem previously assessed for the accumulation and depuration of CTX [13]. An apex meso-predator of this ciguateric food chain is the common coral trout (*Plectropomus leopardus*), one of a group of *Plectropomus* and *Variola* species collectively known as coral trout (grouper) that occasionally cause ciguatera from the GBR [1,8] and is the major target species for the commercial fin-fish fishery on the GBR [9]. The major CTX that is known to contaminate coral trout from the GBR is Pacific-ciguatoxin-1 (P-CTX-1 [16], which is also known as CTX1B) [14]. We developed our model by choosing a target concentration of 0.1 µg/kg of P-CTX-1 (0.1 ppb) in the flesh of a 1.6 kg common coral trout and then back-calculating the quantity of toxin required to be transferred through each trophic level to cause this level of contamination [64]. This model incorporates the production of the less toxic P-CTX-1 precursors P-CTX-4A (CTX4A) and -4B (CTX4B) by *Gambierdiscus* sp. and then the transfer and biotransformation of these in GBR food chains to contaminate common coral trout. “CTX” is used throughout the paper to cover all toxic ciguatoxin precursors and metabolites and is estimated as P-CTX-1 equivalents (eq.).

We modelled scenarios for cell densities of ≤10, 100, and 1000 *Gambierdiscus* epiphytic on turf algae, as well as the minimum *Gambierdiscus* concentration of P-CTX-4A/B that can transfer through food chains to produce a 1.6 kg grouper with a flesh contamination of 0.1 µg/kg P-CTX-1. Calculations for the model variables and assumptions (Appendix A) were performed by using a commercial spreadsheet (Excel). This first attempt to quantify the flow of CTX along coral reef food chains to produce a ciguateric grouper assumes variables scale linearly, providing a starting point for future research to enhance model parameterization, although linear scaling is unlikely in nature. A major limitation of the model is that the uncertainties for the parameters and variables are unknown. It is possible that many variables and their error distributions will differ between species and geographic locations.

## Figures and Tables

**Figure 1 toxins-15-00230-f001:**
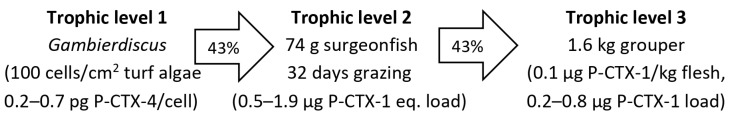
Model for the food chain transfer of P-CTX-4 from *Gambierdiscus* spp. through a ciguateric surgeonfish (detritovore/herbivore) to produce a minimally ciguateric grouper (carnivore) on the Great Barrier Reef (see Section 2.6). Estimated transfer efficiency was 43%, but the longer-term impact of ciguatoxin depuration was not considered in this model.

**Figure 2 toxins-15-00230-f002:**
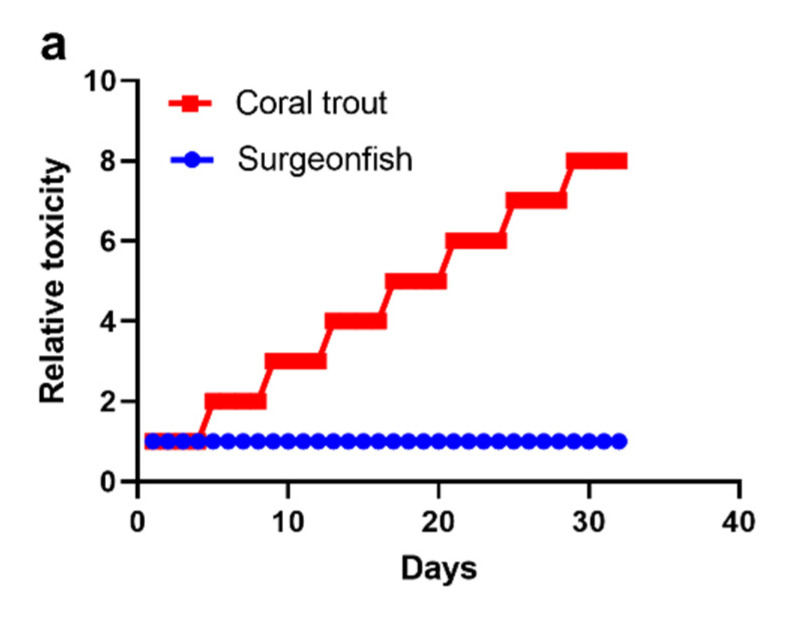
Conceptual model for the change in relative toxicity (arbitrary units) over 32 days of a school of surgeonfish, with 1 fish from the school being eaten by a coral trout every fourth day. The toxicity units are relative within, not between, each species. (**a**) A coral trout feeding on a school of surgeonfish that have accumulated toxicity from ingesting a short-lived bloom of ciguatoxic *Gambierdiscus*, i.e., the relative toxicity of fish in the school of surgeonfish does not change over the 32 days. Data from Appendix A. (**b**) A coral trout feeding from a school of surgeonfish that are accumulating toxicity daily from feeding on a persistent bloom of ciguatoxic *Gambierdiscus*. Data from Appendix A.

**Figure 3 toxins-15-00230-f003:**
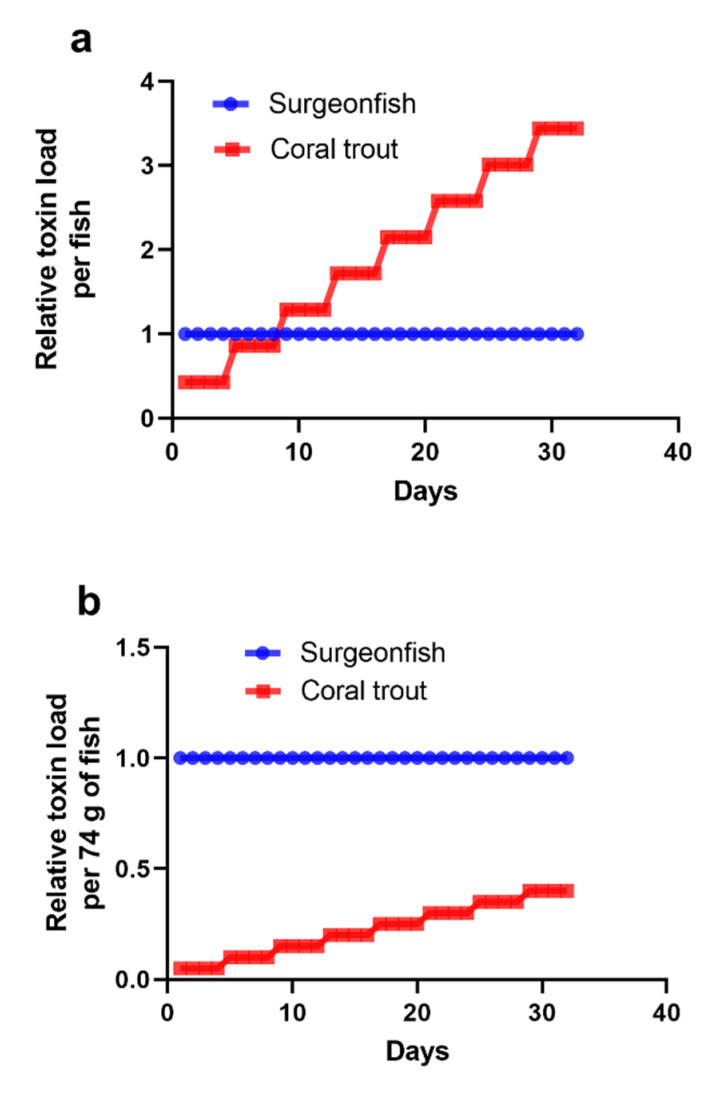
Conceptual model for the change in relative toxicity (arbitrary units) over 32 days of a school of surgeonfish, with 1 fish from the school being eaten by a coral trout every fourth day. (**a**) A coral trout feeding on a school of surgeonfish that have accumulated toxicity from ingesting a short-lived bloom of ciguatoxic *Gambierdiscus* (i.e., Figure 2a with a 43% transfer efficiency across trophic levels [88]). (**b**) Transformation of Figure 3a for 74 g surgeonfish being preyed upon by a 1.6 kg coral trout. Data are expressed on an equivalent weight basis (per 74 g). (**c**) Transformation of Figure 3b for 74 g surgeonfish contaminated with P-CTX-4A being preyed upon by a 1.6 kg coral trout with a 5-fold increase in toxicity across trophic levels, e.g., for P-CTX-4A biotransformed to P-CTX-1 [14,15,37,131].

**Table 1 toxins-15-00230-t001:** Estimated time for 74 g surgeonfish (*C. striatus*) to consume ^1^ 1.1—4.3 µg CTX ^2^ from turf algae supporting 10, 100, or 1000 highly ciguatoxic *Gambierdiscus* producing 1.6 pg P-CTX-4B/cell ^3^ or 0.6 pg of P-CTX-4A and P-CTX-4B/cell ^4^.

P-CTX-4B and/or P-CTX-4A/*Gambierdiscus* Cell (pg/Cell)	Days to Consume Turf Algae with 10 *Gambierdiscus*/cm^2^	Days to Consume Turf Algae with 100 *Gambierdiscus*/cm^2^	Days to Consume Turf Algae with 1000 *Gambierdiscus*/cm^2^
1.6	36–143	4–14	0.4–1.4
0.6	95–382	10–38	1–3.8

^1^ Scenarios based upon 5400 bites over 9 h of feeding/day and consuming 50% of *Gambierdiscus* from a bite of 0.7 sq. cm turf algae. ^2^ Data from Appendix A for accumulation of a CTX load in surgeonfish that, if preyed upon, could produce a 1.6 kg coral trout with a flesh contamination of 0.1 µg/kg P-CTX-1. ^3^ Hypothetical P-CTX-4B concentration [64] and ^4^ combined P-CTX-4A and P-CTX-4B concentrations from *G. polynesiensis* [33].

## Data Availability

Data sources are attributed in the article.

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
