# Peer review of "Model of the Origin of a Ciguatoxic Grouper (Plectropomus leopardus)"

_toxins, 2023, doi:10.3390/toxins15030230_

Round 1

Reviewer 1 Report

Modelling the origin of ciguatoxic grouper (Plectropomus leopardus) on the Great Barrier Reef, Australia

The manuscript is dedicated to model the food chain transfer of CTX from Gambierdiscus to the common coral trout through surgeonfishes. The model is built on previous models. One of the objectives is to estimate the concentration of Gambierdiscus that leads to ciguateric fish on the GBR, the other objective is to identify gaps in the understanding of how ciguateric fishes are produced in the marine food chain. Both objectives are crucial for the management of fisheries and the design of monitoring programs in relation to ciguatera. The manuscript provides conclusions on the densities of Gambierdiscus that will represent a risk or not as well as in complexity of the processes to produce ciguateric fish.

The manuscript is well written, the authors show a good knowledge of the “problem”. Some of the contents in Results should be included in Materials and methods. The section Materials and methods is very short and although the manuscript provides a description of the methodology, the methods are mainly described in the section results and discussion. This may be due to the fact that the journal includes the section Material and methods at the end.

I suggest moving these 2 paragraphs (copied below) from section 2 to section 4, and modify them accordingly considering what is already included in section 4: 

We modelled the hypothetical transfer of ciguatoxins from Gambierdiscus epiphytic on a turf algae layer at surface (plan view) densities of ≤10, 100, and 1,000 cells/cm2 to span the globally reported cell densities from surfaces (mostly screens). We based our model on these cells producing 1.6 x 10-12 g of P-CTX-4B/cell, the same theoretical toxin concentration used for our recent modelling of the contamination of pelagic Spanish mackerel (Scomberomorus commerson) by Gambierdiscus from Platypus Bay on the east coast of Queensland (a non-coral reef habitat south of the GBR) (Holmes and Lewis 2022a).

We modelled the contamination of a 1.6 kg common coral trout (the average size of the commercially harvested fish of this species from the GBR, Leigh et al. 2014) accumulating a target flesh CTX contamination of 0.1 µg/kg P-CTX-1. This concentration is 10-fold higher than the USFDA recommends as the safe limit for consumption of seafood (USFDA 2022) and is the lowest effect level determined by Lehane and Lewis (2000) and formed the basis of the USFDA compliance level for P-CTX. This flesh concentration of CTX would likely produce mild symptoms in 2 out of 10 people (Lehane and Lewis 2000). We calculated a total CTX burden for a 1.6 kg common coral trout using the method of Holmes and Lewis (2022a), based upon a flesh (fillet) recovery weight of 50% total fish weight (Anderson et al. 2003, Sydney Fish Market), and a toxin assimilation efficiency of 43% across each trophic level, with flesh contributing 10%—40% of the total CTX burden (Bennett and Robertson 2021, Holmes and Lewis 2022a)

The section on materials and methods is also missing the type of modelling that was applied.

I suggest organizing the section material and methods in two subsections, 1) data used and 2) modelling strategy and software employed. The manuscript includes a good description about how and why for the data used but not for the modelling strategy. There is a conceptual model of the food chain at the introduction and a good explanation for its simplification.

Other small changes are the following,

Line 40: “Plectropomus spp. and Variola spp.” These 2 are genera. Authors could improve the description of the species complex by listing the species included in the complex.

Line 59 and others: authors should include the taxonomic authority after scientific names the first time that the species that are cited in the text.

Line 632-634: authors should elaborate more about these mechanisms in a schematic or resumed way for the summary.

Author Response

We thank the Reviewer of his/her comments.

Reviewer: The manuscript is well written, the authors show a good knowledge of the “problem”. Some of the contents in Results should be included in Materials and methods. The section Materials and methods is very short and although the manuscript provides a description of the methodology, the methods are mainly described in the section results and discussion. This may be due to the fact that the journal includes the section Material and methods at the end.

Response: We consider the development of the model as part of the Results for the paper so have not changed the format but have incorporated this material under a heading: 2.1 Framework for model development

Reviewer: The section on materials and methods is also missing the type of modelling that was applied. I suggest organizing the section material and methods in two subsections, 1) data used and 2) modelling strategy and software employed. The manuscript includes a good description about how and why for the data used but not for the modelling strategy. There is a conceptual model of the food chain at the introduction and a good explanation for its simplification.

Response: We agree that materials and methods section was too brief and have expanded it with an outline of the modelled scenarios (Lines 643—652) in the Materials and methods section and added 3 tables (S6—S8) to the Supplementary materials listing the variables, values, assumptions, as well as comments and calculations used in the food chain model. As indicated on lines 647, the calculations were done using Excel.

Other small changes

Reviewer: Line 40: “Plectropomus spp. and Variola spp.” These 2 are genera. Authors could improve the description of the species complex by listing the species included in the complex.

Response: Agreed and we have simplified the sentence (Lines 42—45) to: “Coral trout is the local name for groupers belonging to species in the genera…”. In addition, we have added Table S1 to the Supplementary material listing the species of coral trout from the Great Barrier Reef. We originally referred to these as a species complex because some of the species can hybridize.

Reviewer: Line 59 and others: authors should include the taxonomic authority after scientific names the first time that the species that are cited in the text.

Response: Agreed, and this change has been made throughout the manuscript.

Reviewer: Line 632-634: authors should elaborate more about these mechanisms in a schematic or resumed way for the summary.

Response: We are not sure what this comment refers to as the Line numbers referred to correspond to the references section in the original version of the manuscript. No changes to manuscript.

Reviewer 2 Report

See attached.

Author Response

We thank the Reviewer for his/her detailed comments.

Reviewer: Robustness could be improved by using field collected data on Gambierdiscus, at CTX levels in trophic level 2 and 3.

Response: Agreed, but the available data is too limited and not in a form that can compared to the model scenarios. For example, there are no data on Gambierdiscus densities (cells/cm2) from turf algae from the GBR, and the only toxicity data from surgeonfish and coral trout from the GBR were from pooled samples (Lewis and Sellin 1992, Lewis et al. 1994). Our model is a first attempt to synthesize research from disparate research fields on ciguatera, fisheries, and coral reef ecology, to produce quantitative estimates that can be tested by future research, with the variables able to be rearranged to explore different scenarios of interest. We have added a statement to this effect to the Introduction (Lines 25—28). A major limitation of the model is the currently small amount of data available to build and test it against, and the absence of any estimates of uncertainty (as suggested by the Reviewer’s comments).

Reviewer: Title: It would be more accurate to use a title such as “Hypothetical model of Pacific ciguatoxins in the grouper (P. leopardus)” since there seems to be less actual data from the region. At a minimum, “hypothetical” would be a helpful addition for future readers. Is the attribution to the GBR Australia pertinent? It seems that the data used draws from other studies equally and is not specific to the GBR. This is not really a critique, just acknowledgement that the study may have broader implications as written.

Response: We have shortened the title to: “Model of the origin of a ciguatoxic grouper (Plectropomus leopardus)” to indicate the broader potential for the modelling than just to the Great Barrier Reef (GBR). However, we focus on the GBR because of the considerable research available on GBR food chains, fisheries, and coral reef ecology. We have included statements in the Abstract (Lines 25—28), Introduction (Lines 139—140), Results and Discussion (Lines 568—581), and Summary (Lines 623-625) to the effect that the variables can be manipulated to explore alternate scenarios for the accumulation and transfer of CTX through food chains, and that it may be possible to adapt the model to food chains where other families of CTX analogs dominate. However, a confounding feature in many other locations are that the grazers (2nd trophic level) are often harvested, whereas on the GBR they aren’t. This would affect assumptions about the rate of turnover of turf algae. Our approach is to use ciguatera data from a broad range of studies and apply this to a specific scenario, a hypothetical food chain on the GBR. We explicitly state that the model is hypothetical in the Introduction (Line 126), Results and Discussion (Lines 193, 276, 285), and in the Materials and methods (Line 628).

Reviewer: I recommend the authors add some evaluation of the effect of uncertainty in at least a few additional model parameters. This would bolster the conclusions that food web variability needs accounted for when modelling the trophic transfer of CTXs. For example, the model is based on transfer between three trophic levels, but if an additional intermediate organism were added (invertebrate or omnivore), how would this affect the estimation of the Gambierdiscus population at the base of the food web? For comparison to the original estimate using their methods, adding one more trophic link and back calculating using the model start point (0.2-0.8 μg P-CTX-1 eq. in the grouper) would alter the level of CTX consumed in the base of the food chain to be 2.5-10.1 μg P-CTX-1 eq. which equates to 1.6-6.3 million cells consumed (up from 0.7-2.7 million) at a cellular concentration of 1.6 x 10-12 g P-CTX-4B per cell.

Response: As indicated above, the lack of uncertainty around all the variables is a major limitation for the model. This is now acknowledged on Lines 651—652. We agree that adding an additional trophic level greatly changes the population of Gambierdiscus required to produce a toxic fish and acknowledge this possibility in the Introduction (Lines 140-142). Additionally, this work is built upon our previous modelling for the production of a ciguatoxic Spanish mackerel which assumes a four trophic level food chain (Holmes and Lewis 2022a).

Reviewer: Some discussion should be added on how variability of trophic interactions may constrain the outputs of models rather than simply focusing on a highly constrained scenario (as is currently presented). Since the effort in putting the model together has already been achieved, this would/should be quite straightforward.

Response: We consider this as another source of variation and therefore an additional source of unquantified error. We acknowledge in the manuscript (Lines 473—475) that: “While these scenarios for developing toxin-burdens in prey and predator fish help conceptualize processes that lead to ciguateric outbreaks, variations in predator-prey responses and the homogeneity and levels of CTX being transferred produce many possible outcomes.”

Reviewer: Further, manipulating any one of the variables would show how variability in the parameters propagates to the output. Changing a variable will propagate the level of uncertainty in the outputs (e.g., increases the range on number of days to consume 1.1-4.3 μg of total P-CTX-1 eq. in a 74 g surgeonfish). For example, a realistic manipulation could be the bite diameter of surgeonfish. For a smaller surgeonfish, assuming the same feeding rate (bites / day = 5400) and scraping efficiency (50%) as presented in the manuscript, a two-fold reduction in bite area (0.7 -> 0.35 cm2 ) results in half the consumption rate and double the time to reach a consumed burden of 1.1-4.3 μg P-CTX-1 eq.

Response: We have analysed an additional scenario based upon manipulating the variables (Lines 568—581): 2.6 The minimum CTX concentration in Gambierdiscus that could produce a ciguateric grouper. We have also added statements in the Abstract (Lines 26—28), Introduction (Lines 139—140), Results and Discussion (Lines 579—581), and Summary (Lines 623-625) to the effect that the variables can be manipulated to explore alternate scenarios for the accumulation and transfer of CTX through food chains, and that it may be possible to adapt our model to food chains where other families of CTX analogs dominate. However, we are cautious in analysing too many scenarios given previous criticism of our modelling as being too speculative. Also, as we acknowledge in Lines 647—649, this first attempt to quantify the flow of CTX along coral reef food chains assumes that the variables scale linearly, which is unlikely. However, it is a starting point for future research to improve the model parameterization. A major limitation of the model is that the uncertainties for the parameters and variables are unknown. It is possible that many variables and their error distributions will differ between species and geographic locations.

Explicit points that need to be addressed (apologies if some overlap with those above):

Reviewer: 1. All of the assumptions of the model should be provided in a supplementary table (or methods).

Response: Agreed, and these are provided in Tables S6—S8 in the Supplementary material.

Reviewer: 2. What are the explicit assumptions in this simplified 3-level food chain model? All formulas used in the model should be provided in the manuscript methods (or as a supp table) so that others can benefit from the work. This could be deduced to produce the table provided above, but took some effort, and might be missing something deemed critical to the assumptions.

Response: Agreed, and these are provided in Tables S6—S8 in the Supplementary material.

Reviewer: 3. What would changing variables do to the calculations? Two have been outlined above, but there are several more (e.g., halving or doubling the size of the grouper). Manipulating some of the variables and reporting on the effect would enhance the results/discussion.

Response: We have analysed an additional scenario 2.6 The minimum CTX concentration in Gambierdiscus that could produce a ciguateric grouper. As all parameters scale linearly in the model it is easy to estimate outputs based upon halving or doubling values. We acknowledged that linear scaling is unlikely in nature (Line 649).However, it is a starting point for future research to improve model parameterization.

Reviewer: 4. Are these scenarios realistic? What are the caveats and limitations? Please add a little more to the discussion on this.

Response: This is difficult question to answer. We clearly state in the manuscript that this is an analysis of a hypothetical food chain. Our model attempts to analyse the flow of CTX from source to creation of a ciguateric fish that would cause human poisoning. It produces quantitative estimates that can be tested against future research. We have included an analysis of an additional scenario (Lines 586—581) 2.6 The minimum CTX concentration in Gambierdiscus that could produce a ciguateric grouper which suggests that the model can produce realistic outputs.

Reviewer: 5. The model doesn’t quite meet the stated goals (both in the intro and discussion) so should be revised a little

Response: We have modified the Introduction (Lines 126—140) to better reflect the manuscript.

Reviewer: 6. The calculations are mathematically correct, but the conceptual design could be improved using data collected or available from the specified region of interest.

Response: The manuscript is based extensively on data from the GBR for trophic levels 2 and 3. There is much better data on the toxicity of Gambierdiscus from other regions, which we incorporate where possible. The lack of regionally specific data does make it difficult to analyse the flow of CTX along an entire food chain.

Reviewer: 7. Figure 1 is a little crude and given how simple the model is, could be (and is) already clearly stated in the text. As provided, this doesn’t quite meet the expectation of a “conceptual model” in modeling terms. I would suggest removing or improving this schematic. If the parameters tested are simple but the discussion moves beyond these two arrows, then the readers would benefit from your perspective on how various other factors could impact the results (e.g., dashed arrows for unknowns, solid for known etc.)

Response: We agree that Figure 1 is a simple (basic) conceptual model, but we prefer to retain it in the manuscript.

Reviewer: 8. There is quite a lot of unnecessary discussion on 24-h screens that detract from the study. The focus of the manuscript would be improved by deleting text from line 143-189 since screens were not evaluated here. The authors describe the toxin source as Gambierdiscus cells/cm turf algae, so just justify the cells/area selected and move on e.g., 10-100 cells/cm2 reflect 24-h screen data from other regions while 1000 cells/cm is more in line with turf/macrophyte counts etc. A minor (2-3 lines) discussion on the challenge of cell/surface area measurements from 3D surfaces would suffice.

Response: We disagree as, at present, the screen assays offer the only estimates for Gambierdiscus cell densities based on surface area. No change to manuscript.

Reviewer 3 Report

This article summarizes the modeling of trophic transfer of ciguatoxins to coral trout in the GBR based on known data of highly ciguatoxic Gambierdiscus, feeding behaviors of herbivorous fish and theoretical densities of ciguatoxic Gambierdiscus.

Comments:

The authors should explicitly state that the modeled estimates are based on consumption of only highly ciguatoxic Gambierdiscus cells, but that in nature it is probable that on any surface the distribution of different species and/or ranges in toxicity may occur.

Ln 62-63: this statement needs to be clarified. It states the highest cellular production of CTXs is reported in G polynesiensis and G excentricus, but care should be made when stating this, given that CTXs have not been confirmed in G. excentricus, but rather this is based on bioactivity reported as "toxin cell quotas” and ciguatoxin-like activity.

Data is presented in ug when discussing fish contamination, while in cells it is in ×10-12 g/cell, it would be easier to follow if the authors modified the units for the cells  to pg/cell.

CTX4A/B concentration estimate – The authors should clearly state that the concentration used for their estimates of highly toxic microalgae was determined by LC-MS, given that many authors use toxin cell quotas estimated by bioasssays (which have limitations in estimating toxin content)

ln 154 = corelated = correlated

L306 = remove "yet"

Author Response

We thank the Reviewer for his/her comments

Reviewer: The authors should explicitly state that the modeled estimates are based on consumption of only highly ciguatoxic Gambierdiscus cells, but that in nature it is probable that on any surface the distribution of different species and/or ranges in toxicity may occur.

Response: Agreed. This is stated (and also inferred) throughout the manuscript. Lines 199—201 state: “This CTX concentration is 2.7-fold greater than the maximum known combined cellular concentrations of P-CTX-4A and -4B of 0.6 pg/cell produced by cultured G. polynesiensis (Darius et al. 2022).” It is also inferred in Lines 266—271, and in the new added section (Lines 568—581): 2.6 The minimum CTX concentration in Gambierdiscus that could produce a ciguateric grouper. We have also added the sentence (Lines 206—207: “However, it is likely that turf algae often host a mix of Gambierdiscus species and/or strains containing a range of ciguatoxin concentrations.”

Reviewer: Ln 62-63: this statement needs to be clarified. It states the highest cellular production of CTXs is reported in G polynesiensis and G excentricus, but care should be made when stating this, given that CTXs have not been confirmed in G. excentricus, but rather this is based on bioactivity reported as "toxin cell quotas” and ciguatoxin-like activity.

Response: Agreed, and we have changed the sentence to: (Lines 65—68) “Globally, the highest cellular production of ciguatoxins has been reported from G. polynesiensis Chinain et Faust 1999 by HPLC-MS/MS and cell-based and immuno-assays, and G. excentricus Fraga 2011 by cell-based and immuno-assays (Chinain et al. 2010a, Fraga et al. 2011, Pisapia et al. 2017, Litaker et al. 2017, Gaiani et al. 2020) but these species have not yet been identified from the GBR”

Reviewer: Data is presented in ug when discussing fish contamination, while in cells it is in ×10-12 g/cell, it would be easier to follow if the authors modified the units for the cells to pg/cell.

Response: Agreed, and we have changed the units to “pg/cell” throughout the manuscript.

Reviewer: CTX4A/B concentration estimate – The authors should clearly state that the concentration used for their estimates of highly toxic microalgae was determined by LC-MS, given that many authors use toxin cell quotas estimated by bioasssays (which have limitations in estimating toxin content)

Response: Agreed, and Lines 65—68 incorporate this addition.

Reviewer: ln 154 = corelated = correlated

Response: Agreed and spelling corrected.

Reviewer: L306 = remove "yet"

Response: Agreed, and the word has been deleted.